# Methionine Modulates the Growth and Development of Heat-Stressed Dermal Papilla Cells via the Wnt/β-Catenin Signaling Pathway

**DOI:** 10.3390/ijms26041495

**Published:** 2025-02-11

**Authors:** Shu Li, Xiaosong Wang, Gongyan Liu, Fuchang Li

**Affiliations:** 1Key Laboratory of Efficient Utilization of Non-Grain Feed Resources (Co-Construction by Ministry and Province), Shandong Provincial Key Laboratory of Animal Nutrition and Efficient Feeding, College of Animal Science and Technology, Shandong Agricultural University, Tai’an 271017, China; 2021010083@sdau.edu.cn (S.L.); 14768123432@163.com (X.W.); 2Shandong Academy of Agricultural Sciences Institute of Animal Husbandry and Veterinary Medicine, Jinan 250100, China; gongyanliu@foxmail.com

**Keywords:** methionine, heat stress, dermal papilla cells, Wnt, rabbit

## Abstract

This study furnishes insights into how methionine mitigates heat-stress-induced impairments in hair follicle development in Rex rabbits at the cellular level. Dermal papilla cells from the dorsal skin of Rex rabbits were isolated, cultured in vitro, and divided into six groups, i.e., control (37 °C; 0 mM methionine), heat stress (45 °C; 0 mM methionine), and heat stress + methionine (45 °C; 15 mM, 30 mM, 45 mM, and 60 mM methionine), with six replicates per group. The heat stress groups were exposed to 45 °C, 5% CO_2_, and 95% humidity for 30 min, followed by recovery at 37 °C, repeated three times over three days. On the third day, samples were collected post-heat stress. The results show that methionine markedly fortified HSP70, MSRA, and SOD expression (*p* < 0.01); augmented proliferation (*p* < 0.01); ameliorated cell cycle progression; and lessened apoptosis (*p* < 0.05). Adding Wnt signaling pathway activators and inhibitors manifested that these effects were associated with diminished β-catenin phosphorylation and aggrandized expression of the Wnt10b, β-catenin (*p* < 0.001), and LEF/TCF nuclear transcription factors (*p* < 0.01). Thus, this study demonstrates that methionine regulates the growth and development of heat-stressed hair papilla cells via the Wnt signaling pathway, remitting heat-stress trauma.

## 1. Introduction

Climate change is affecting increases in global temperature, with the last decade marked by unprecedented heatwaves that are proceeding in a chronic trend [1], resulting in severe consequences for biological systems, particularly at the cellular level. Hyperthermia suppresses essential cellular processes such as DNA repair, leading to cytotoxicity, protein denaturation, enzyme inactivation, mitotic obstruction, and cell death (apoptosis, pyrosis, ferroptosis, etc.). These alterations accrete with temperature [2]. In eukaryotes, cytoskeletal defects are a significant consequence of heat stress. The cytoskeleton renders structural support and is crucial for multiple cellular functions. Heat stress also perturbs RNA splicing, disrupts membrane mobility [3], and expedites cell aging [4]. The main traits of cell senescence are irreversible growth stagnation and the appearance of aging-correlated phenotypes, mainly exerted as heightened activeness of selenium-interrelated β-galactosidase (SA-β-Gal) and cell cycle arrest [5,6]. Sustaining protein balance is essential for withstanding stress and ensuring healthy aging [7]. The primary cytoplasmic response to heat stress is the transcriptional activation and accumulation of highly conserved heat shock proteins (HSPs) [8], which act as endocellular molecular chaperones that endow livestock heat endurance to disparate stressors, reputed to subside fulfilling biomarkers for appraising heat stress plentitude in livestock [9]. Heat stress decreased hair follicle density and fur quality; it induced the apoptosis and inhibited the proliferation of dermal papilla cells [10].

DPCs are clusters of mesenchymal cells at the bases of hair follicles; they are deemed specialized mesenchymal stem cells [8,11]; they interact with epithelial precursor cells in early embryonic development to drive epithelial cell proliferation, substrate formation, and hair follicle downward growth, ultimately inducing hair follicle formation. After hair follicle maturation, DPCs continue to secrete factors through WNT/BMP/SHH/FGF and other signaling pathways to regulate the growth cycle, color, thickness, and type of hair [12]. Certain biomarkers have been employed to discern DPCs, such as alpha-smooth muscle actin (α-SMA), alkaline phosphatase (ALP), and vimentin (Vim). DPCs are the signaling center of the hair follicle; the number of DPCs influences the size and shape of the hair through diverse mechanisms [13]. The subcutaneous injection of DPCs can promote hair follicle regeneration and hair growth [14]. During the hair follicle enlargement stage, DPCs motivate stem cells with different signals and initiate the cell growth stage. A rise in Wnt, stabilization of β-catenin, and inhibition of BMP are signs of the inception of the growth phase [15].

The proliferation and differentiation of DPCs are mainly regulated through classical Wnt conduction [16]. The Wnt signaling pathway is evolutionarily conserved in mammals and represents one of the foremost signal transduction pathways governing hair follicle growth, development, and cycles [17]. It has been attested that the activation of the Wnt/β-catenin signaling pathway conduces hair follicle regeneration [18]. The endoplasmic reticulum synthesizes Wnt ligands and secretes them as glycoproteins, which bind to frizzled receptors on the plasmalemma. The intracellular destruction complex, consisting of glycogen synthase kinase (GSK-3β), adenomatous polyposis coli (APC), Axin2, and casein kinase (CK-1), is restrained by the scaffold protein disheveled (DVL), which accelerates the accumulation of β-catenin in the cytoplasm and its subsequent translocation into the nucleus to interact with T cell/lymphocyte enhancer transcription factor (TCF/LEF), promoting the transcription of Wnt target genes such as *c-MYC* and *CyclinD1* [19].

The sequencing of discrepantly expressed genes in primary and secondary hair follicles revealed that these genes were mainly involved in the metabolism of methionine and cystine [20]. Methionine (Met) is the starting amino acid for protein synthesis in all known organisms, the foundation of protein translation, a precursor for cell methylation, and a ligand in REDOX metalloproteins [21,22,23]. Methionine sulfoxide represents the dominant product of methionine oxidation and may also function as an inhibitor of methionine metabolism [24]. Methionine sulfoxide reductases (MSRs) are pivotal enzymes that reduce oxidized Met residues in proteins to restore their function, and they have strict stereospecificity for their substrates [25,26]. Met deficiency can hasten cell senescence and disrupt cell signaling and protein function [27,28]. Met suppression can intensify the cytotoxicity induced by heat stress [29].

In our previous study, methionine subsided the scathe of hair follicle development in Rex rabbits caused by heat stress [30], but the underlying mechanism requires further exploration. We speculate that Met ameliorates the growth and development of heat-stressed dermal papilla cells through the Wnt signaling pathway, thereby impacting hair follicle development. The outcomes of this study demonstrate that Met regulates the growth and development of thermally stressed dermal papilla cells via the Wnt signaling pathway. This study provides a reference for the study of nutrition’s effects on hair follicle growth and development under environmental stress.

## 2. Results

### 2.1. Cell Morphological Identification

As shown in Figure 1, Giemsa staining expounded that the cells were long spindle-shaped and exhibited swirl-like aggregative growth characteristics. The positivity rates for α-SMA and VIM immunofluorescence staining exceeded 98%, indicating that the detached cells were dermal papillae.

### 2.2. Impacts of Methionine on Intracellular Enzyme Concentrations and Cell Viability

As illustrated in Figure 2A–F, in comparison with the control group, heat stress signally reduced the ALP, MARA, and SOD levels in dermal papilla cells and their proliferation, and it tended to decrease CAT activity. The addition of 15 mM and 30 mM Met notably enhanced the ALP concentration; 15 mM and 45 mM Met addition significantly boosted cell proliferation, and methionine addition strikingly increased MSRA, SOD, and CAT activity. As shown in Figure 2B, heat stress remarkably increased the intracellular HSP70 concentration, but methionine supplementation had no apparent effect on this. Figure 2C indicates that neither heat stress nor methionine affected intracellular SAM levels.

### 2.3. Effects of Methionine on Protein Levels in Heat-Stressed Hair Papilla Cells

As depicted in Figure 3, heat stress markedly increased the p-β-catenin level and notably weakened Wnt10b and β-catenin. Met markedly increased Wnt10b and β-catenin levels; 15 mM and 30 mM Met inhibited the phosphorylation β-catenin, while 45 mM and 60 mM Met did not show stronger effects.

Based on the above data, 30 mM Met was selected for the subsequent experiments.

### 2.4. Effects of Methionine on Cell Senescence and Apoptosis

As Figure 4 shows, β-galactosidase staining demonstrated that heat stress particularly increased the level of β-galactosidase compared with that in the control group, and the cells showed enlarged and flattened morphological variations (Figure 4A). The addition of methionine reduced the β-galactosidase activity. Flow cytometry showed that, confronted with the control group, heat stress reduced the proportion of G0/G1 phase cells but increased the ratio of cells in the S phase (Figure 4B), early apoptosis, and total apoptosis (Figure 4C). Methionine addition reversed these effects.

### 2.5. Effects of Methionine on Gene Expression in Heat-Stressed Hair Papilla Cells

As shown in Figure 5, heat stress markedly increased the gene expression of *BMP2*, *Hes5*, and *FGF5* compared to the control group, and it decreased the gene expression of *HGF*, *FGF7*, *VCAN*, *Wnt10b*, *β-catenin*, *TCF*, *LRP6*, and *DVL2*. Conversely, the gene expression of *BMP2*, *Hes5*, and *FGF5* was conspicuously decreased, while *HGF*, *Wnt10b*, *β-catenin*, *TCF*, and *LRP6* were notedly cemented and returned towards normal upon the addition of methionine.

### 2.6. Effects of Wnt Pathway Activators and Inhibitors on the Activity of Related Enzymes

As shown in Figure 6, compared with the heat stress group, SKL2001 markedly increased ALP, SOD, and MSRA activity and *HSP70* gene expression in heat stress cells. In contrast to that in the heat stress + methionine group, XAV939 also noteworthily inhibited the activity of ALP, SOD, and MSRA and the gene expression of *ALP*, but it had no notable effects on the content and gene expression of *HSP70*.

### 2.7. Reflections of Wnt Pathway Activators and Inhibitors on Cell Senescence and Apoptosis

As expounded in Figure 7, compared with the heat stress group, SKL2001 addition notably reduced β-galactoside staining (Figure 7A) and the proportion of cells in early apoptotic and total apoptotic states (Figure 7B), and it tended to increase *Bcl2/Bax* gene expression (Figure 7D). Compared with the heat stress + methionine group, XAV939 brilliantly augmented the rate of β-galactosidase staining (Figure 7A), late apoptotic and total apoptotic cells (Figure 7B), and *Caspase3* gene expression (Figure 7D), while cell proliferation activity was notably reduced (Figure 7C).

### 2.8. Effects of Wnt Pathway Activators and Inhibitors on Gene and Protein Expression

As elaborated in Figure 8, relative to the heat stress group, SKL2001 singularly intensified *LRP6* and *LEF1* expression, modulated the gene expression of *β-catenin* and *Wnt10b* (Figure 8A) and the levels of the HSP70 and Wnt10b proteins, and reduced β-catenin phosphorylation (Figure 8B). Compared to the heat stress + methionine group, XAV939 extraordinarily decreased *Wnt10b* and *LEF1* expression and tended to reduce *β-catenin* expression (Figure 8A), while it outstandingly increased the HSP70 and P-β-catenin protein levels and reduced the Wnt10b protein level (Figure 8B).

## 3. Discussion

Evolution has optimized the metabolic processes of various species to accommodate dissimilar body temperatures, thereby complicating the interpretation of developmental responses to heat. Due to the substantial variation in the normal body temperatures of various species, it is impractical to set a single heat stress temperature for all [31]. The rectal temperature of mice is approximately 36.5 ± 1.3 °C, with the heat stress model temperature set at 43 °C [32]. The body temperature of camels is around 40.7 °C, and the temperature of the oocyte heat stress model is 45 °C for 2 h [33]. The human skin fibroblast heat stress model temperature is 45 °C for 30 min [34]. Based on previous reports, the rectal temperature of rabbits is about 39.1 ± 0.5 °C; hence, the heat stress model established in this experiment used a temperature of 45 °C, with a continuous temperature for 30 min and a long-term stress of 3d.

Protein quality control, also known as protein deposition, regularized protein synthesis, folding, unfolding, and turnover by chaperone proteins, protease systems, and cellular clearance mechanisms such as autophagy and lysosomal degradation [35]. Hsp70 is central to protein deposition, coordinating cellular functions by directing substrates to fold, depolymerize, refold, or degrade, thereby powering cellular resistance to stress [36]. Heat stress is an evolutionarily conserved response, and HSP70 is its biomarker. Stable ALP activity is beneficial for maintaining DPCs’ follicle-inducing ability [37]. In this study, the α-SMA- and Vim-positive rates were higher than 98%, and HSP70 expression was increased during the high-temperature treatment, indicating that the heat stress model of dermal papilla cells was successfully constructed in vitro. Based on this model, adding 30 mM methionine especially increased the activity of ALP, promoted the proliferation of DPCs, and protected the development of the hair follicle center, suggesting that methionine has a certain antagonistic effect on heat stress and can ameliorate the function and development of DPCs under heat stress.

Heat stress leads to the accumulation of toxic compounds, including reactive oxygen species, thus triggering oxidative stress, activating the antioxidant defense system in the body, and increasing the levels of antioxidant enzymes [38]. In this experiment, the SOD, CAT, and MSRA levels in the heat stress group all notably increased. Methionine residues are the sole residues that can be recovered via the oxidative modification of sulfur-containing amino acids. The sulfur group can inactivate free radicals, while the cyclic REDOX of Met residues is a critical antioxidant mechanism [39]. Individuals lacking the *MSRA* gene are more susceptive to oxidative stress [40]. Methionine replenishers can raise antioxidant capacity through the direct or indirect glutathione system and the thioredoxin–MSRA pathway [28,41]. Methionine increased the MSRA, SOD, and CAT activity, reinforced the antioxidant capacity through the MSRA pathway, and restored REDOX homeostasis.

All organisms must adopt strategies to cope with environmental stresses that can disrupt cell structure and function or even lead to cell death if left unchecked. While organisms can eliminate impaired cells through apoptosis, the excessive activation of apoptosis by environmental fluctuations must be avoided [42]. Cell death is manipulated by both anti-apoptotic and pro-apoptotic factors; the pro-apoptotic factor *Bax* oligomerizes and forms channels in the outer membranes of the mitochondria of stressed cells, facilitating the delivery of pro-apoptotic factors, thus activating caspase3 and promoting protein degradation. The anti-apoptotic BCL2 family, on the other hand, directly controls cell fate by physically blocking the oligomerization of Bax [43]. The apoptosis of cells under heat stress is achieved to some extent through the mobilization of Bax mediated by NoXa-dependent MCL1 protein loss, induced by HSP70 [44]. Heat stress can induce cell cycle arrest by causing G0/G1 or G2/M phase arrest, by inducing temporary stagnation (i.e., arrest at cell cycle checkpoints) during G1/S and G2/M phase transitions, or by promoting early cell senescence in time of the S phase [45,46]. Methionine deficiency reduces the expression of cycle-regulating molecules, and the inhibition of methionine metabolism leads to cell senescence and cycle arrest [47]. These findings imply that methionine could abate the cell senescence, S phase arrest, and apoptosis induced by heat stress.

Wnt signaling is a comprehensive regulator of cellular protein catabolic metabolism [48]. Transcriptomic studies of chicken hepatocellular carcinoma cell lines suggested that the cumulative effect of heat stress on the TGF and WNT pathways is mediated through the inhibition of apoptosis (TGF inhibition) and proliferation (WNT inhibition), promoting cell survival [49]. Epidermal WNT/β-catenin activation promotes the transition from the telogen phase to the growth phase [50]. Sustained activation of WNT/β-catenin in various epidermal or dermal stem cells generates different types and degrees of ectopic hair follicles, with different effects on skin and hair homeostasis [51]. The addition of Wnt pathway inhibitors notably mitigated the symptoms of hyperalgesia in rats [19]. The experiments also showed that heat stress disrupted the early stage of DPC DNA synthesis, diminished cell proliferation, increased cell apoptosis, and increased the expression of genes and proteins related to the Wnt and TGFβ signaling pathways. Methionine was required for Wnt-induced endocytosis, and methionine exhaustion blocked Wnt signaling, which was reversed by adding SAM to the culture medium [48]. Methionine facilitated feather growth in chicks by actuating the Wnt/beta-catenin signaling pathway [52]. Preliminary results also showed that methionine may reduce the disruption of hair follicle development in heat-stressed Rex rabbits through the Wnt signaling pathway [32]. In this study, by adding Wnt signaling pathway agonists and inhibitors, it was confirmed that methionine can promote the growth and development of Rex rabbit hair papilla cells under heat stress via the Wnt signaling pathway. These findings stress the importance of the Wnt signaling pathway in the cellular response to heat stress and also highlight the role of methionine as a key molecule regulating this process, providing new perspectives for understanding the molecular mechanisms of animal cell growth and development under heat stress conditions.

It is essential to acknowledge the limitations of this work. This experiment did not further knock out or knock down the Wnt pathway gene; doing so could more thoroughly elucidate the mechanisms and also create opportunities for further exploration in subsequent experiments.

## 4. Materials and Methods

### 4.1. Isolation, Culture, and Identification of Primary Dermal Papilla Cells

The method for isolating and identifying Rex rabbit hair papilla cells was drawn from Liu et al. [53], without sex discrimination. Briefly, clipped skin was shredded into approximately 1 cm^2^ lumps, immersed in a double volume of mixture including 0.25 mg/mL dispase II (Sigma, Livonia, MI, USA) and 1% penicillin–streptomycin (NCM, Suzhou, China) at 4 °C overnight to potentiate the separation of the epidermis from the dermis, and rewarmed at 37 °C for 30 min the next day; it was then digested in a solution subsuming 0.1 mg/mL of collagenase D (Sigma, Livonia, MI, USA) and 1% penicillin–streptomycin for 5–6 h. The cell suspension was centrifuged with gradient rotation and percolated through a 75 mM filter to remove debris. The filtered cells were inoculated in Dulbecco’s Modified Eagle Medium (DMEM, Gibco, New York, NY, USA) supplemented with 10% fetal bovine serum (FBS, Gibco, New York, NY, USA) and 1% penicillin–streptomycin, cultured in a 37 °C incubator with 5% CO_2_ and 100% humidity, and purified through trypsin digestion. The cell purity was evaluated through an immunofluorescence assay with α-SMA and Vim as specific markers, and only cells with 98% purity were utilized.

### 4.2. Experimental Design

Dermal papilla primary cells were cultivated and passaged in cell culture dishes; once the cells reached 80% confluence, the original medium was discarded and replaced with DMEM (Gibco, New York, NY, USA) without Met and serum for overnight assimilation. The cells were stochastically divided into 6 groups, i.e., a control group (37 °C; 0 mM Met), heat stress group (HS, 45 °C; 0 mM Met), and heat stress + methionine group (45 °C; 15 mM Met), heat stress + methionine group (45 °C; 30 mM Met), heat stress + methionine group (45 °C; 45 mM Met), heat stress + methionine group (45 °C; 60 mM Met), with 6 replicates per group. The heat stress groups were placed in a 45 °C, 5% CO_2_, 100% saturated humidity incubator for half an hour and then reheated in a 37 °C incubator [34] three times over three days at once. On the third day, the samples were gathered after heat stress.

### 4.3. Cell Proliferation and Viability

The cell concentration was adjusted to 10^4^/mL. Cells at 100 μL per well were inoculated into 96-well cell culture plates and cultured/treated as described above. After heat stress on the third day, the cells received new mediums and were substituted and incubated at 37 °C for 1 h with the addition of 10 μL of CCK-8; the absorbance at 450 nm was then measured.

### 4.4. Correlative Enzymes Activity and Protein Levels

The cells were enumerated following digestion and then seeded into 6-well plates at a density of 2 × 10^5^ cells per well; samples were taken after the culture treatment as previously described. The intracellular HSP70, SAM, and MSRA contents were measured using an enzyme-linked immunosorbent kit (Enzyme-linked Biotechnology, Shanghai, China), while the activities of intracellular alkaline phosphatase ALP, SOD, and CAT were determined using a microplate assay (Jiancheng, Nanjing, China). The specific procedures were performed according to the kit’s instructions.

A Western blot was conducted to quantify the relative levels of proteins associated with the pathway. Specifically, cells were lysed and ultrasonically broken in RIPA buffer with a protease phosphatase inhibitor and PMSF (NCM Biotech, Suzhou, China). Afterward, they were centrifuged at 12,000 rpm at 4 °C for 10 min, the supernatant was collected, the protein concentration was determined using a BCA protein quantitative kit (CWBIO, Suzhou, China), and the protein was denatured with sodium dodecyl sulfate (SDS) (NCM Biotech, Suzhou, China) at 100 °C. Next, the proteins were separated into bands through electrophoresis in a Bis-tris polyacrylamide gel (e-life, Beijing, China); the proteins were then transferred to a methanol-activated polyvinylidene fluoride film (Millipore, Billerica, MA, USA) in a transmembrane buffer containing 10% anhydrous ethanol, at 400 mA for 30 min at room temperature. The membrane was sealed using a rapid blocking buffer (NCM Biotech, Suzhou, China) at room temperature for 10 min; it was then incubated with a primary antibody overnight at 4 °C with gentle shaking. The primary antibodies used were anti-Wnt10b, anti-β-catenin (Cell Signaling Technology, Trask Lane Danvers, MA, USA), anti-HSP70 (Proteintech, Wuhan, China), and anti-p-β-catenin (Abclonal, Wuhan, China). The membrane was then washed three times with 1 × TBS-Tween buffer (NCM Biotech, Suzhou, China) and incubated with relevant rabbit or mouse secondary antibodies at 4 °C for 3–4 h. After washing, the membrane was imaged using ECL supersensitive luminescent solution (affinity, Suzhou, China) and quantitatively analyzed in Fusion FX VILBER LOURMAT (VIBER GmbH, Paris, France).

Based on the results of the above assays, a methionine concentration of 30 mM was selected for the subsequent experiments.

### 4.5. Cell Senescence

Trypsin-digested dermal papilla cells were uniformly spread on 6-well culture plates. Following group treatment, the medium was discarded, and the cells were washed twice with PBS. Then, 1 mL of β-galactosidase stain and fixing and working buffer (Beyotime, Shanghai, China) were added, and observations were made using an ordinary optical microscope.

### 4.6. Flow Cytometry

Fourth-generation dermal papilla cells, digested using trypsin, were evenly distributed on 6-well culture plates. After treatment, the medium was removed, and the cells were digested with trypsin without EDTA. Apoptosis was determined using the Annexin V-FITC/PI Apoptosis Detection Kit (BD, New York, NY, USA) program. The specific steps of the cell cycle detection were as follows: Cells were placed into 1.5 mL tubes, rinsed with PBS, fixed using 70% cold ethanol at −20 °C for 18 h, and centrifuged at 500 rpm for 5 min. Then, 0.5 mL of PI/RNase staining solution was added after the ethanol had been absorbed. The cells were mixed thoroughly, incubated at 37 °C for 15 min, and stored at 4 °C in the dark. Flow cytometry was completed within 1 h. ModFit LT5.0 software was used to analyze the cell ratio of each sample period.

### 4.7. Quantitative Real-Time RT-PCR

Total RNA was extracted from the cells using Trizol and reverse transcribed into cDNA with a reverse transcription kit (EG241025, Yugong Biotech, Lianyun gang, China) for real-time quantitative polymerase chain reaction. The 260/280 ratios of all the samples were determined using a nucleic acid spectrophotometer (DENOvix, Wilmington, DE, USA) to ensure that they were greater than 1.8 and less than 2.1. The integrity of the RNA was verified through 0.6% agarose gel electrophoresis and staining with ethidium bromide (EB, Sigma, Livonia, MI, USA). The housekeeping gene *GAPDH* was used for normalization, and the 2^−ΔΔCT^ method [54] was used to calculate the relative gene expression. All the primers used for qRT-PCR are displayed in Table 1.

### 4.8. Identification of Molecular Mechanisms

To further investigate the role of the Wnt10b/β-catenin signaling pathway in methionine’s influence on DPC development under heat stress, the cells were categorized into five groups: control group, HS group (45 °C; 0 mM Met), HS + SKL2001 [55,56] (Wnt signaling pathway agonist) group (45 °C; 0 mM Met), HS+ methionine group (45 °C; 30 mM Met), and HS + methionine +XAV939 [57,58] (Wnt signaling pathway inhibitor) group (45 °C; 30 mM Met). The treatment and determination analysis were performed in the same manner as described above.

### 4.9. Statistical Analysis

All the experiments were repeated at least three times (n = 3). The data are presented as the mean ± standard error. Using SAS9.4 software, one-way analysis of variance (ANOVA) was performed for each group to determine significant differences, followed by Duncan’s multiple range test (three or more groups) or a *t*-test (two groups). Statistical significance was determined using Prism 9.5 (GraphPad Software, San Diego, CA, USA).

## 5. Conclusions

Heat stress reduced the proliferation and antioxidant enzyme levels of DPCs, precipitated the senescence and apoptosis process, and induced cell cycle arrest. Methionine inhibited these effects by upregulating the Wnt signaling pathway, thereby protecting the growth and development of DPCs under heat stress. Further studies are needed to determine which specific genes play key roles.

## Figures and Tables

**Figure 1 ijms-26-01495-f001:**
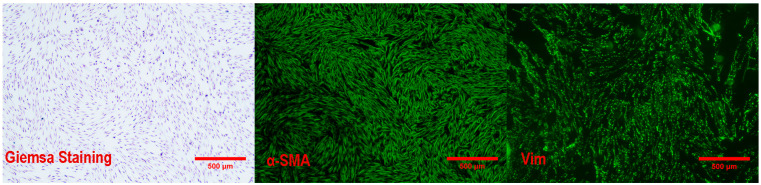
Identification of primary cell markers (10 × 4).

**Figure 2 ijms-26-01495-f002:**
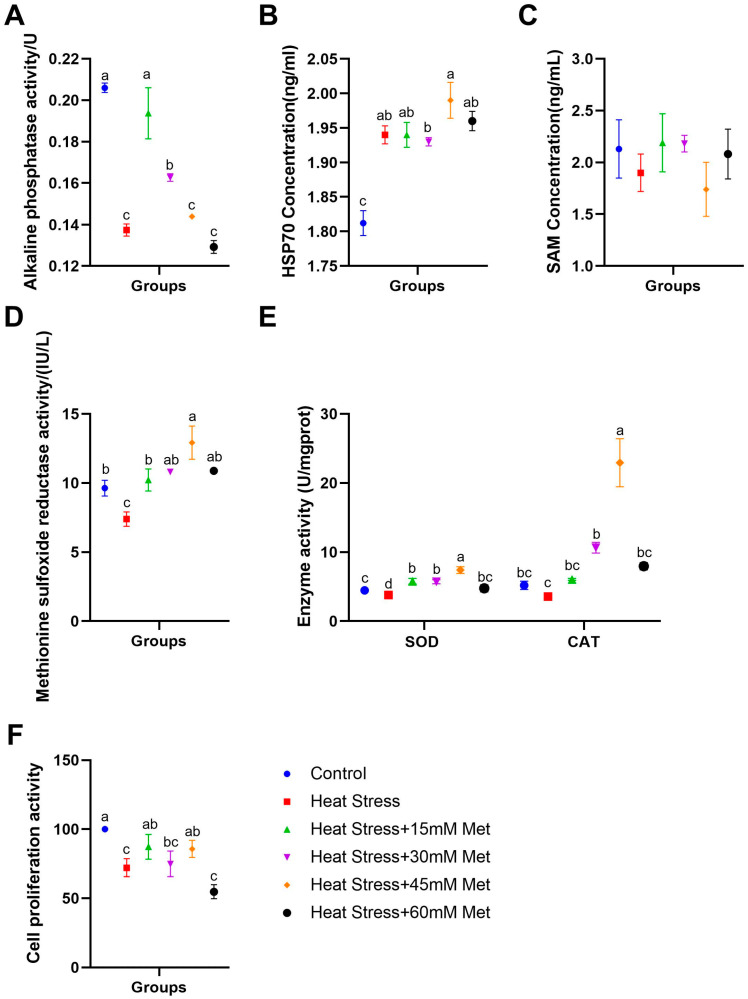
Impacts of methionine on intracellular enzyme concentrations and cell viability. (**A**) ALP activity. (**B**) HSP70 content. (**C**) SAM assay. (**D**) MSRA level. (**E**) SOD and CAT activity. (**F**) Cell proliferation activity. Means with different superscript letters are statistically significantly different (*p* < 0.05).

**Figure 3 ijms-26-01495-f003:**
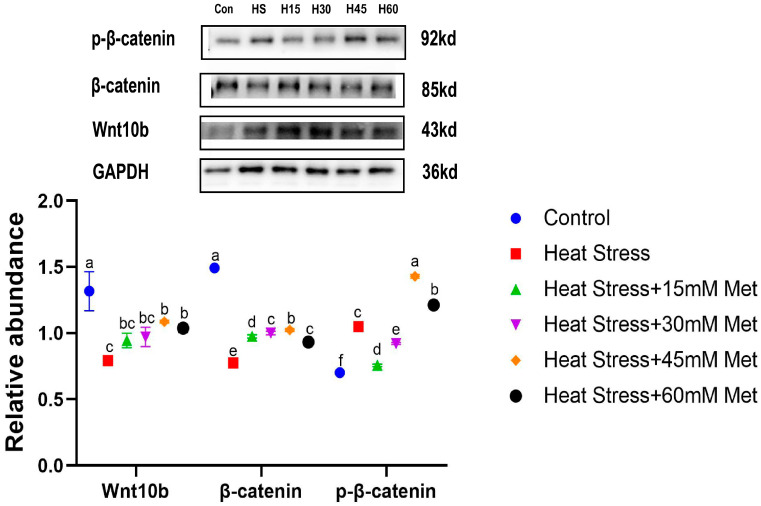
Effects of methionine on protein levels in heat-stressed hair papilla cells. Means with different superscript letters are statistically significant (*p* < 0.05).

**Figure 4 ijms-26-01495-f004:**
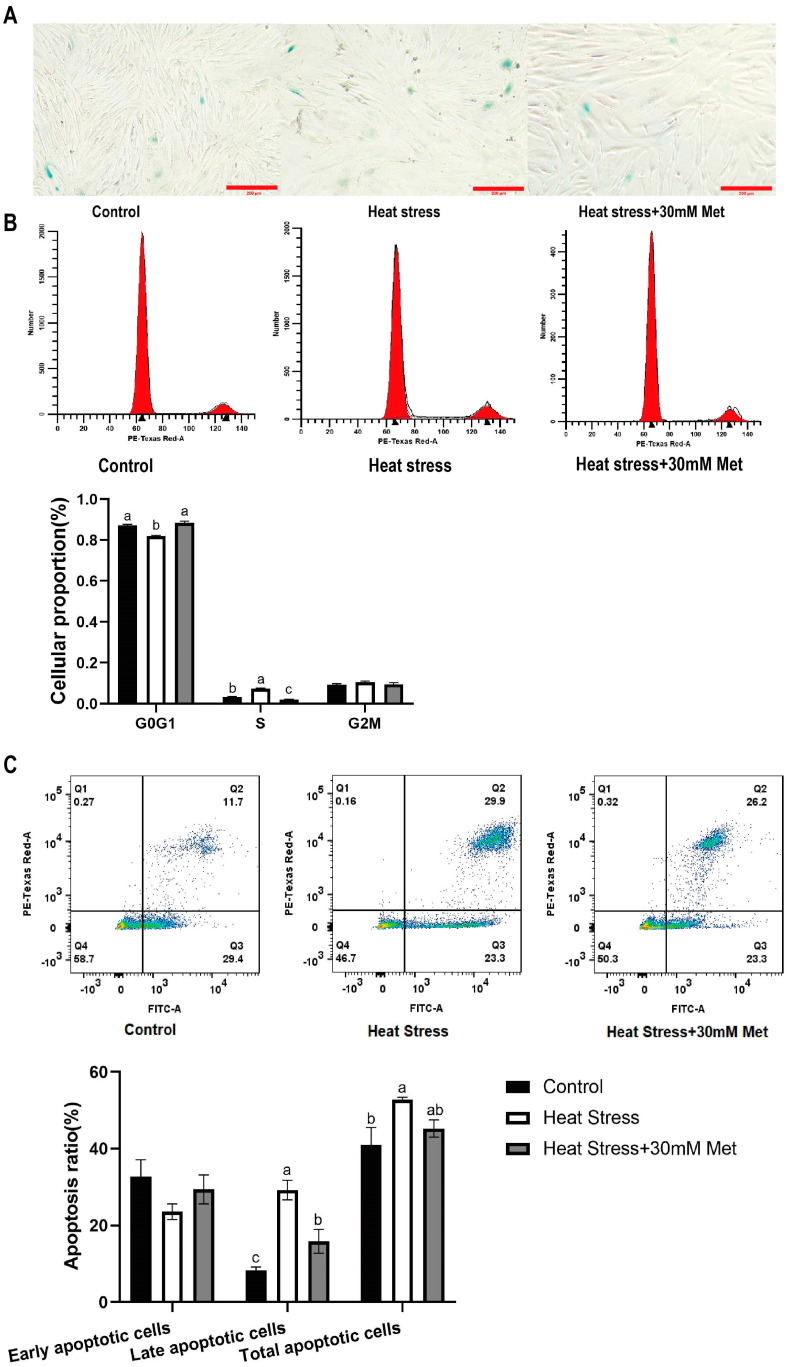
Effects of methionine on cell senescence and apoptosis. (**A**) Expression of Cellular senescence state marker (10 × 4). (**B**) Effect on the cell cycle. (**C**) Effect on cell apoptosis (Q1: Cell debris or mechanically damaged cells; Q2: Late apoptotic cells; Q3: Early apoptotic cells; Q4: Normal cells). Means with different superscript letters are statistically significant (*p* < 0.05).

**Figure 5 ijms-26-01495-f005:**
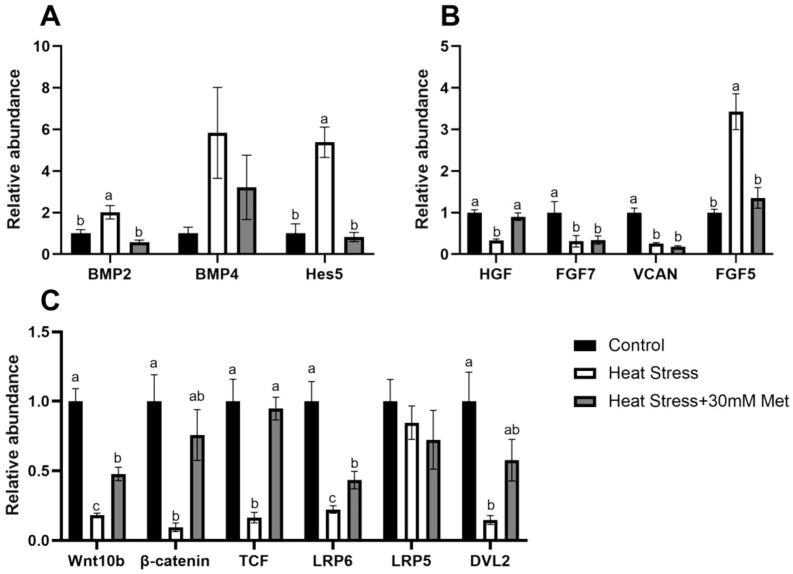
Effects of methionine on gene expression in heat-stressed hair papilla cells. (**A**) Major BMP signaling pathway genes. (**B**) Related factors. (**C**) Major genes of the Wnt signaling pathway. Means with different superscript letters are statistically significantly different (*p* < 0.05).

**Figure 6 ijms-26-01495-f006:**
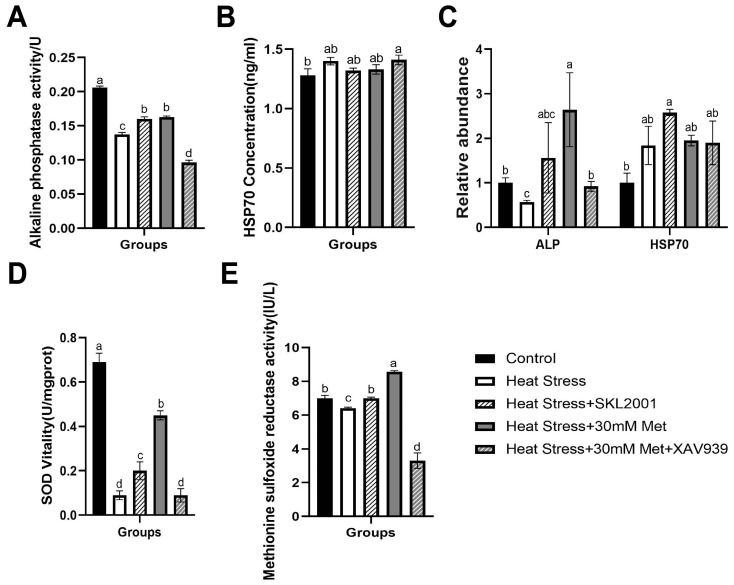
Effects of Wnt pathway activators and inhibitors on the activity of related enzymes. (**A**) ALP activity. (**B**) HSP70 content. (**C**) Gene expression of ALP and HSP70. (**D**) SOD activity. (**E**) MSRA level. Means with different superscript letters are statistically significantly different (*p* < 0.05).

**Figure 7 ijms-26-01495-f007:**
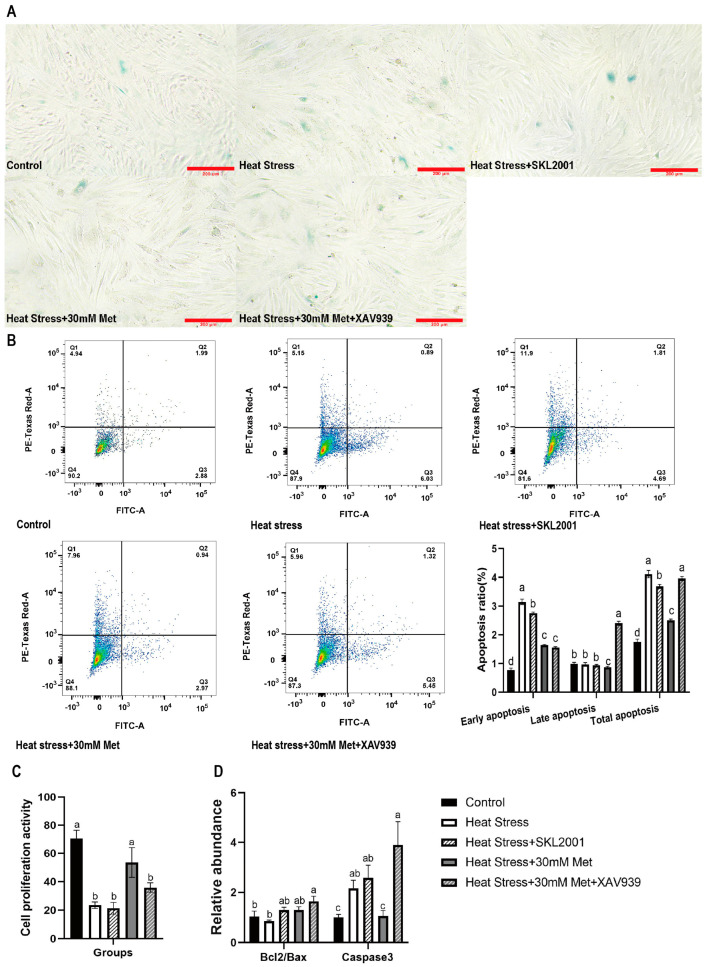
Effects of Wnt pathway activators and inhibitors on cell senescence and apoptosis. (**A**) β-galactose staining (10 × 4). (**B**) Apoptosis ratio (Q1: Cell debris or mechanically damaged cells; Q2: Late apoptotic cells; Q3: Early apoptotic cells; Q4: Normal cells). (**C**) Cell proliferation activity (**D**) Expression of apoptosis-related genes. Means with different superscript letters are statistically significantly different (*p* < 0.05).

**Figure 8 ijms-26-01495-f008:**
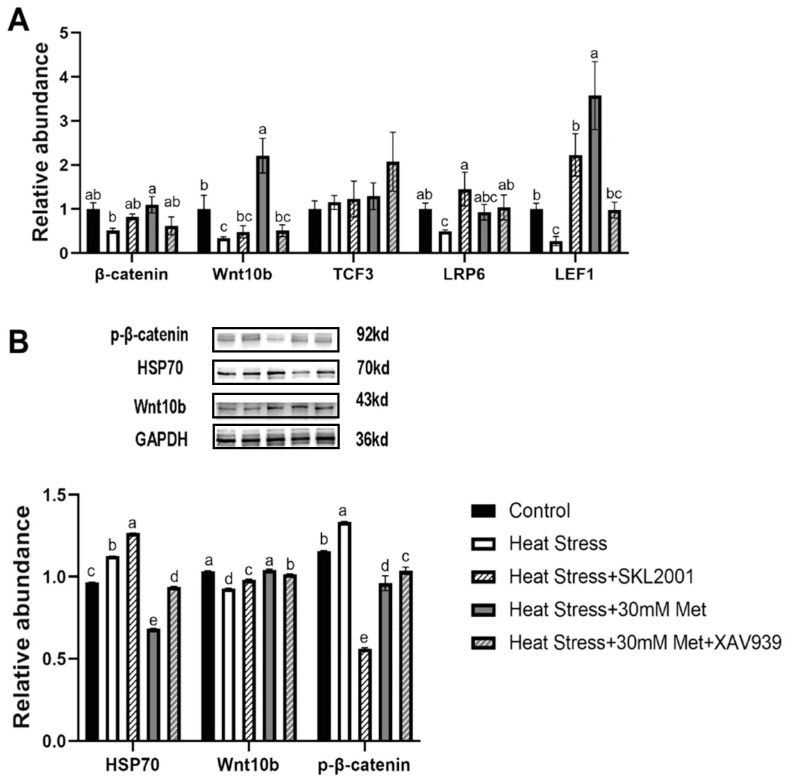
Effects of Wnt pathway activators and inhibitors on gene expression and protein levels. (**A**) Expression of major Wnt signaling pathway genes. (**B**) Levels of major Wnt signaling pathway proteins. Means with different superscript letters are statistically significantly different (*p* < 0.05).

**Table 1 ijms-26-01495-t001:** Specific gene primer sequence.

Genes	GenBank Accession No.	Primer Sequence (5′-3′)	Product Size/bp
*LRP5*	AB017499.1	F: CCTTTACGAGCGGAACCAC R:GCAGGGTAGAACACGTCCAT	144
*Hes5*	NM_008253710	F:AGACCGCATCAACAGCAGCATC R:ATCTCCAGGATGTCCGCCTTCTC	105
*VCAN*	XM_008261907.3	F:AATCATCCCTTTAGTCACCG R:CCTGTGGATAATCTGTCTGG	126
*DVL2*	XM_008270807	F:ACTCCACCATGTCCCTCAA R:CGATGTAGATGCCTCCGTCT	117

Note: Wnt10b, HGF, β-catenin, BMP2, BMP4, FGF5, FGF7, TCF, LRP6 and LEF1 primer sequences refer to the previous (Li et al., 2022 [30]).

## Data Availability

The data supporting the findings of this study are available on request from the corresponding author.

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
