# Peer review of "Methionine Modulates the Growth and Development of Heat-Stressed Dermal Papilla Cells via the Wnt/β-Catenin Signaling Pathway"

_ijms, 2025, doi:10.3390/ijms26041495_

Round 1

Reviewer 1 Report

Comments and Suggestions for Authors

The paper is about  the protective role of methionine against heat stress in hair follicle development in Rex rabbits. By isolating and culturing dermal papilla cells under controlled conditions, the researchers effectively created a model to assess the impact of increased temperatures on cellular health. The results highlight that methionine not only enhances the expression of essential protective proteins like HSP70 and SOD but also promotes cell proliferation and improves cell cycle progression while reducing apoptosis. Moreover, the study elucidates the underlying mechanisms by linking these protective effects to the Wnt signaling pathway. This connection opens avenues for further research into nutritional interventions that could stabilize cellular functions during environmental stressors, not only in animal models but potentially extending this knowledge to human applications. Overall, this work contributes valuable insights into stress response in hair follicle biology and the potential use of methionine as a mitigative agent.

Revisions suggested: 

- In the introduction section there is not a clear link between Heat-stress consequence and DP cells. I suggest to add a sentence linking the two concept.

-Line 53-57 are redundant and not usefull for the scope of the paper

- Name of genes must be in italic

- Conclusion section must be improved with reference to genes and proteins up-regulated and down-regulated.

-Limitations of thes tudy must be included

-I suggest also to improve the future perspective of the findings related to hair growth and WNT pathway stimulation

Comments on the Quality of English Language

The English could be improved to more clearly express the researc especially in the Results and  Discussion section 

Author Response

Dear Reviewer,

Thanks very much for taking your time to review this manu. I really appreciate all your comments and suggestions! Please find my itemized responses in below and my revisions/corrections in the re-submitted files.

Comment 1. In the introduction section there is not a clear link between Heat-stress consequence and DP cells. I suggest to add a sentence linking the two concept.

Reply 1. Thank you for pointing out this. I agree with this comment. A sentence is added in lines 46-47 to connect heat stress and DP.

Comment 2. Line 53-57 are redundant and not useful for the scope of the paper.

Reply 2. Agree. Removes redundant in lines 53-57.

Comment 3. Name of genes must be in italic.

Reply 3. Agree. Unitalicized gene names have been corrected and highlighted in red.

Comment 4. Conclusion section must be improved with reference to genes and proteins up-regulated and down-regulated.

Reply 4. Agree. The conclusion section was improved based on the up-regulation and down-regulation of gene proteins.

Comment 5. Limitations of the study must be included.

Reply 5. Agree. Limitations of the study have been added in the discussion section.

Comment 6. I suggest also to improve the future perspective of the findings related to hair growth and WNT pathway stimulation.

Reply 6. Thanks for your suggestion. Subsequent experiments will focus on finding methionine targets and knock-out or knock-down validation.

Comments on the Quality of English Language

The English could be improved to more clearly express the research especially in the Results and Discussion section

Reply. The paper has been polished using the MDPI language editing system.

Reviewer 2 Report

Comments and Suggestions for Authors

This study investigates how methionine alleviates heat stress-induced damage to rabbit dermal papilla cells by activating the Wnt/β-catenin signaling pathway. The findings reveal that methionine significantly enhances the activity of antioxidant enzymes, reduces cell apoptosis, improves cell proliferation, and regulates the expression of genes and proteins related to the Wnt signaling pathway. These results suggest that methionine plays a crucial role in maintaining hair follicle cell function and coping with heat stress, providing new insights into hair follicle development and environmental stress regulation.

I would like to suggest the following revisions to the manuscript:

  1. The scale bar and numerical markers in Figure 1 are too small and unclear, making them difficult to interpret. They should be adjusted for better visibility.
  2. The Western blot images in Figure 3 could be appropriately enlarged to enhance their clarity and readability.
  3. In line 239, it is mentioned that heat stress affects the Wnt signaling pathway by influencing apoptosis and modulating Wnt signaling through its impact on cell proliferation. However, it should also be noted that the Wnt signaling pathway continuously influences dermal and dermal papilla cells under normal conditions (e.g., Y. Gao, The FASEB Journal, 2024; J. Han, European Journal of Dermatology, 2023). It would be beneficial to include references discussing the effects of Wnt/β-catenin on dermal papilla cells under normal conditions.

Overall, this is a very intriguing article. I hope the authors can address the aforementioned points to further improve the manuscript.

Author Response

Dear Reviewer,

Thanks very much for taking your time to review this manu. I really appreciate all your comments and suggestions! Please find my itemized responses in below and my revisions/corrections in the re-submitted files.

Comment 1. The scale bar and numerical markers in Figure 1 are too small and unclear, making them difficult to interpret. They should be adjusted for better visibility.

Reply 1. Thank you for pointing out this. I agree with this comment. The scale bars and markers of Figure 1 have been corrected to make them clearer.

Comment 2. The Western blot images in Figure 3 could be appropriately enlarged to enhance their clarity and readability.

Reply 2. Agree. The protein bands in Figure 3 are enlarged to make them more clearly visible.

Comment 3. In line 239, it is mentioned that heat stress affects the Wnt signaling pathway by influencing apoptosis and modulating Wnt signaling through its impact on cell proliferation. However, it should also be noted that the Wnt signaling pathway continuously influences dermal and dermal papilla cells under normal conditions (e.g., Y. Gao, The FASEB Journal, 2024; J. Han, European Journal of Dermatology, 2023). It would be beneficial to include references discussing the effects of Wnt/β-catenin on dermal papilla cells under normal conditions.

Reply 3. Agree. The effect of Wnt/β-catenin on dermal papilla cells under normal conditions has been discussed, and all recommended references are cited.
